# Primary Care Physicians’ Perceptions on Nurses’ Shared Responsibility for Quality of Patient Care: A Survey

**DOI:** 10.3390/ijerph191710730

**Published:** 2022-08-29

**Authors:** Yael Sela, Tamar Artom, Bruce Rosen, Rachel Nissanholtz-Gannot

**Affiliations:** 1Nursing Science Department, Faculty of Social and Community Sciences, Ruppin Academic Center, Emeq-Hefer 4025000, Israel; 2Smokler Center for Health Policy Research, Myers-JDC-Brookdale Institute, Jerusalem 9103702, Israel; 3Department of Health Systems Management, School of Health Sciences, Ariel University, Ariel 4076414, Israel

**Keywords:** community health services, nurses, physicians, primary care, quality of health care

## Abstract

Nurses are key players in primary care in Israel and in the efforts to improve its quality, yet a survey conducted among primary care physicians (PCPs) in 2010 indicated that 40% perceived the contribution of nurses to primary care quality as moderate to very small. In 2020, we conducted a cross-sectional survey using self-report questionnaires among PCPs employed by health plans to examine the change in PCPs’ perceptions on nurses’ responsibility and contributions to quality of primary care between 2010 and 2020. Four-hundred-and-fifty respondents completed the questionnaire in 2020, as compared to 605 respondents in 2010. The proportion of PCPs who perceive that nurses share the responsibility for improving the quality of medical care increased from 74% in 2010 to 83% in 2020 (*p* < 0.01). Older age, males, self-employment status, and board certification in family medicine independently predicted reduced PCP perception regarding nurses’ responsibility for quality-of-care. PCPs who believed that nurses contribute to quality of practice were 7.2 times more likely to perceive that nurses share the responsibility for quality-of-care. The study showed that over the past decade there was an increase in the extent to which PCPs perceive nurses as significant partners in improving quality of primary care.

## 1. Introduction

Many countries have adopted quality measurement programs [1]. In Israel, the National Program for Quality Indicators in Community Healthcare (hereafter the National Program), which was established in 2000, collects data on 70 indicators, including data on preventive, diagnostic, and rehabilitative care provided in the community and furnishes information to policymakers and the public [2].

Israel has a national health insurance system that provides universal coverage to all citizens and permanent residents of Israel. Each individual freely chooses from four competing, not-for-profit health plans (also referred to in the literature as health maintenance organizations, sick funds, or health funds). These health plans provide their members with access to a statutory benefits package. The healthcare system is financed by general taxes and an earmarked payroll tax (health tax). These funds are allocated to the health plans according to a capitation formula, with risk adjustment intended to sufficiently compensate the plans for the cost of members’ care. The members in each health plan choose their primary and specialist community-based physicians from physicians affiliated with the health plan [3,4]. The health plans vary in size, the characteristics of their members in terms of income, age and geographic location, the extent of reliance on salaried versus self-employed physicians, the extent of reliance on group practices versus solo practitioners, and whether the plan owns acute care hospitals [5].

In recent decades, the workload in community primary care has increased due to the rise in chronic illnesses and the growth of the elderly population [6]. Most quality indicators in community healthcare relate to areas that are the responsibility of primary care physicians (PCPs), adding to their perceived workload [1]. One way to cope with PCPs’ increased workload and to improve the quality-of-care is to provide primary care by a multidisciplinary team [7,8,9,10,11,12].

Nurses play a key role in multidisciplinary teams in primary care [13,14,15] and as the mainstay of the contact between patients and physicians [16,17]. In many countries the development of nurses’ roles and their involvement in care have led to the expansion of their authority and training [14,18]. A scoping review of literature relating to registered nurses’ role in primary care/public health collaboration found that the roles of nurses included care coordinator, program facilitator, outreach professional, and relationship builder. Furthermore, registered nurses supported transitions in chronic disease, communicable disease care, and maternity care at various healthcare system levels including systemic, organizational, intrapersonal, and interpersonal levels [19]. Nurses also have a role in health promotion and prevention [20]. Some studies have shown that nursing care can serve as an economical substitute to a physician’s management of patient care without compromising quality in the case of minor health problems [21,22].

Family physicians and nurses differently perceive the nursing role in primary care. Whereas nurses noted the clinical value, physicians tended to identify their role as “supportive” despite their broad training and authority to perform complex treatments [15]. A study conducted in Israel in 2010 on the attitude of PCPs towards the contribution of nurses to improving quality-of-care found that 40% of respondents assessed the contribution to be moderate or small. Moreover, some 25% of respondents felt that the responsibility of nurses for improving quality-of-care was small/very small [1].

Since 2010, the role of community nurses in Israel has changed. Specifically, nurses moved from reactive to initiated work, were given more authority and responsibility in the management of chronically ill patients, a more central role in health promotion efforts, more advanced training, and more opportunities to focus on roles and tasks that require nursing professionals [23]. Considering these changes, the goal of this study was to examine whether PCPs’ perceptions on nurses’ shared responsibility and contribution to quality-of-care in the framework of the National Program has changed since 2010. Our findings will help in establishing policies to enhance the acceptance and appreciation for nurses’ contribution to quality-of-care, particularly among PCPs, and in the long-run should contribute to attracting individuals to the nursing profession, especially in light of the ever-growing shortage of nurses [4,24,25].

## 2. Materials and Methods

### 2.1. Setting and Study Population

Two cross-sectional surveys were conducted, in: 2010 and 2020, among representative samples of PCPs working in four national public health plans. The results of the 2010 survey were published previously [1,2].

The study population consisted of PCPs working for the health plans (full- or part-time, salaried or self-employed contractors), engaged in the direct care of adult patients. Paediatricians, physicians with no responsibility for the quality of care for a panel of patients: consultants, physicians engaged mainly in administrative or managerial work, retired physicians, and temporary replacements were excluded from the study population. The study team estimated that approximately 4400 Israeli physicians met these criteria. In each survey, a random stratified sample of 1000 PCPs was selected from the administrative records of the health plans—250 from each of the four health plans. In the 2010 survey, 884 PCPs met the eligibility criteria and were asked to participate in the survey, thereof 605 responded and completed the questionnaire (a response rate 68.4%). In the 2020 survey, 725 PCPs met the eligibility criteria, thereof, 450 completed the questionnaire (a response rate 62.0%). The main reasons for non-response were refusal to participate in the study and difficulty in making contact (18% and 20%, respectively, for the 2020 survey).

The study was approved by the institutional ethics committee (approval number IRB-BH-261, approval date 20 November 2018). All participants provided their consent for participating in the study and were assured anonymity.

### 2.2. Instrument

The development of the study questionnaire was previously described [1]. Briefly, the study questionnaire comprised items referring to the PCPs’ perceptions on the extent of nurses’ shared responsibility for improving the quality-of-care, and the extent of the actual contribution of their involvement to the quality of practice. The participants were asked to rate their responses on a scale of 1 (very little or not at all) to 6 (to a very large extent). The questionnaire consisted of identical items.

Key dependent variables were PCPs’ perception on the extent of nurses’ shared responsibility for improving healthcare quality, and perception of nurses’ actual involvement in the quality of practice.

The independent variables included PCPs’ demographic and professional characteristics. Other covariates were perceived projection of patients’ psycho-social state on the medical condition and success of treatment, the level of responsibility and sense of added burden due to follow-up on quality indicators, and efforts to improve them.

### 2.3. Data Collection

Data were collected from May 2019 to January 2020. All PCPs in the sample were sent an email invitation to participate in the survey, a link to the web-based questionnaire. The PCPs could complete the instrument online, or alternatively, could respond by telephone, regular post, or fax. Designated respondents received up to four reminders by telephone or email. The mix of response modes in 2020 differed from that of 2010. In 2020, 78.2% of the respondents chose to complete the instrument online, while in 2010 only 14.5% completed the questionnaire online and 85.5% had completed it by regular post, telephone, or fax.

### 2.4. Data Analysis

Analyses were conducted using the Statistical Package for the Social Sciences (SPSS), version 24 (IBM, Armonk, NY, USA).

Differences in study populations’ characteristics between 2010 and 2020, and bivariate analysis of the association of the independent variables with PCPs’ perceptions on nurses’ shared responsibility for improving the quality-of-care, were examined by study year by the chi-squared test. Logistic regression models were performed to assess the extent to which the participants’ characteristics and perceptions contribute to their perception of nurses’ shared responsibility for quality-of-care. The data were weighted to reflect the differences among the health plans in sampling ratios and response rates (health plan-specific response rates ranged from 59% to 67%) so that the results would more accurately reflect the national study population. The weighting also considered the relationship between the sampling probability and the number of health plans where each physician worked (i.e., a physician working for two health plans was more likely to be included in the sample than a physician working for only one plan). All statistical analyses were conducted by using the weights.

## 3. Results

### 3.1. Respondent Characteristics

The characteristics of the respondents to both surveys are summarized and compared in Table 1. At both time points there were no statistically significant differences in gender, board certification, and employment status. Approximately half of respondents were 45–60 years old, but in 2020 a greater percentage of respondents were over 60 years of age than in 2010 (33% vs. 19%). In both years, most respondents were Jewish (76% in 2010 and 71% in 2020), were not born in Israel (60% in 2010 and 51% in 2020), and practiced primary care (92% in 2010 and 88% in 2020). The differences between years reflect the changing processes of the survey’s target population in terms of aging and an increase in the proportion of experts.

### 3.2. PCPs’ Perceptions on Nurses’ Shared Responsibility for the Improvement of Quality-of-Care

The percentage of PCPs who perceived that health plan nurses share the responsibility for improving the quality-of-care to a great/very-great extent increased from 74% in 2010 to 83% in 2020 (*p* = 0.019) (Figure 1). Only a minority of PCPs perceived that health plan nurses do not share this responsibility or share it to a very small extent (9% in 2010 and 8% in 2020).

### 3.3. PCPs’ Perceptions on the Contribution of Nurses’ Actual Involvement to the Quality of Practice

As shown in Table 2, in 2020, two-thirds of PCPs (67%) perceived that nurses’ actual involvement contributes to the quality of practice to a great/very great extent (*p* < 0.05). The proportion of respondents who did not perceive that nurses make a substantial contribution to the quality-of-care declined from 41% in 2010 to 33% in 2020 (*p* < 0.05). The main change was observed in the percentage of PCPs who perceived that the nurses’ involvement in practice contributes to quality-of-care to a very great extent (an increase from 17% in 2010 to 25% in 2020; *p* < 0.05).

Bivariate analyses of PCPs who perceived that nurses share the responsibility for quality-of-care to a very great extent (Table 3) indicated that a higher proportion of respondents in 2020 than in 2010 believed that psycho-social determinants affect patients’ medical condition and treatment success (52% vs. 43%); devoted less than 5% of their time to follow-up and improving quality indicators (48% vs. 34%); agreed that they share the responsibility for patient care with nurses (70% vs. 63%); and that they are responsible for the performance of some quality-of-care indicators while nurses are responsible for others (64% vs. 56%). An increase over the years was also noted for the sense of added burden due to follow-up on indicators (42% in 2010 and 46% in 2020). About half of the PCPs who perceived that the psycho-social state projects on the medical condition and the success of treatment to a very great extent, also believed that nurses share responsibility to a very great extent.

### 3.4. Independent Predictors of PCPs’ Perceptions Regarding Nurses’ Responsibility for the Improvement of Quality-of-Care

Logistic regression of data of PCPs who perceived that nurses share the responsibility for quality-of-care to a very great extent showed that older age, self-employed status, and board certification in family medicine were predictors of lower perception regarding the extent of nurses’ responsibility for quality-of-care, while Israeli-born and Jewish PCPs were more likely to perceive nurses as sharing responsibility to a very great extent (Table 4). Furthermore, PCPs who believed that nurses actually contribute to the quality of practice to a very great extent were also 7.2 times more likely to perceive that nurses share the responsibility to a very great extent compared to PCPs who believed that nurses actually contribute to the quality of practice to a lesser extent. Finally, PCPs who participated in the 2020 study were 1.2 more likely to perceive that nurses share the responsibility to a very great extent than PCPs who participated in the 2010 survey.

## 4. Discussion

Our findings show that between 2010 and 2020 the percentage of PCPs perceiving that nurses share responsibility to a great extent for improving the quality-of-care increased. Good interprofessional collaboration has been shown to improve health outcomes and patient safety [26,27], the quality of patient care [28], and to reduce patient mortality [29]. Physicians’ perceived view of collaboration and communication with nurses depends on the work environment, the organization, country of origin, and culture. For example, physicians working in Norwegian hospitals perceived that they had effective collaboration, particularly if they believed that they were the controlling partner in nurse–physician collaborations [30]. In another study, physicians working in outpatient oncology clinics in Ontario, Canada reported that nurses’ input on patient care was extremely valuable [31]. Italian primary care paediatricians had a positive opinion about having a paediatric nurse in their office and rated ‘very useful’ most of the suggested activities that paediatric nurses could perform in their offices, specifically, healthcare education and disease prevention [32]. Lack of physician recognition of nurses’ professional roles was found to be a major obstacle for establishing good communication and collaboration between these professions [33,34,35].

In the past 20 years, Israeli health plans have regularly reviewed the quality of preventive, diagnostic, and rehabilitative services that they deliver. The findings of the quality monitoring program are publicized and are fully transparent, showing comparative data on health plans; hence, quality indicators lead to intra- and inter-organizational competition. Furthermore, all Israeli health plans have initiated steps to promote community health and regard nurses as the most suitable professional workforce for the task since they combine clinical knowledge with an appropriate approach to patients [16]. These changes may have led physicians to consider nurses as greater contributors to quality-of-care than in the past.

Our results also show that PCPs 45 years and older were less likely to perceive that nurses share the responsibility and contribute to quality-of-care compared to those younger than 45 years. These findings are in line with those reported by Gualano et al. [16], who also showed that physicians’ age is an important predictor of how essential they perceived a nurse’s presence alongside the family physician: older physicians (>50) perceived joint nurse-physician work as less important. Similarly, Dall’Oglio found that younger paediatricians who collaborated with a secretary and other workers in the office considered the activity of nurses providing healthcare education in their office as more useful compared to older participants [32]. While primary care is still based largely on the physician-focused traditional model of care, younger physicians did exhibit a positive attitude towards a model of enhanced sharing, which may be a promising starting point. Matziou et al. [33] also reported that younger physicians with less clinical experience recognized nurses’ administrative skills and tended to accept their opinion about treatment and decision making more easily.

According to the psycho-social approach, treatment does not consist solely of biological clinical indicators, but also includes aspects of psychological and social care [26]. About half of PCPs who believed to a very great extent that the psycho-social state affects patients’ medical condition and treatment success, also believed that nurses share responsibility for improving quality-of-care to a very great extent. Gualano et al. [16] noted that physicians’ perceptions on nurses’ contribution to improving quality-of-care focus on their interpersonal skills. Physicians tend to acknowledge the contribution of nurses in their “traditional” role of patient support in psycho-social aspects rather than clinical ones. It appears reasonable, therefore, that physicians who attribute importance to the psycho-social aspects of quality-of-care also attribute considerable importance to nurses’ involvement nurses in care.

PCPs’ workload has increased in recent years and is expected to grow with the rise in complex morbidity. PCPs are required to summon, counsel, and sometimes also treat and follow-up patients who were not actively monitored in the past. In a report published in 2013, the Israel Medical Association contended that the heavy workload of PCPs has shortened clinic visits to 5 min, in which it is impossible to deliver high-quality care [3]. In the current study, we found that PCPs who felt that quality monitoring increased their workload to a very great extent were more likely than other PCPs to perceive that nurses contribute to their practice to a great/very great extent. According to a study conducted in Israel, teamwork should be introduced to enhance physicians’ sense of meaning in work and job satisfaction, and to improve quality and patient satisfaction in order to deal with physician burnout due to overload [36]. Therefore, it may be assumed that if PCPs feel that the level of care provided would not be inferior to their own, various aspects of care, such as follow-up on chronic patients, could be transferred to another party. Thus, as their workload increases, PCPs perceive nurses as more important, and nurses assume greater responsibility for various time-consuming aspects of care.

Although some of the demographic parameters collected in 2020 were different from those of the 2010 survey (age, birth origin, profession, occupation), they were adjusted for the previous results. The survey was done among a representative stratified random sampling of PCPs working in each of the four national public health plans; therefore, its findings represent the perceptions of Israeli PCPs working in the country’s public health system. The two surveys were conducted a decade apart, and although the questions about nurses’ contributions were identical, there was a difference in the method for completing the questionnaires: In the 2010 survey, 82% of questionnaires were completed by regular mail and in 2020 most questionnaires were completed online. These different means for survey completion may have led to a social desirability bias among respondents, despite the assurance of anonymity.

## 5. Conclusions

Over the past decade, more PCPs perceive nurses as significant partners in improving quality-of-care. This may be partly attributed to the growing shortages in staff and other resources, coupled with PCPs’ desire to provide appropriate health services in line with improving quality-of-care measurements, which make it essential for them to share their workload and responsibilities with nurses. This constitutes an opportunity for nurses to expand their role and may lead to enhanced acceptance and appreciation for nurses’ contribution to quality-of-care by PCPs.

## Figures and Tables

**Figure 1 ijerph-19-10730-f001:**
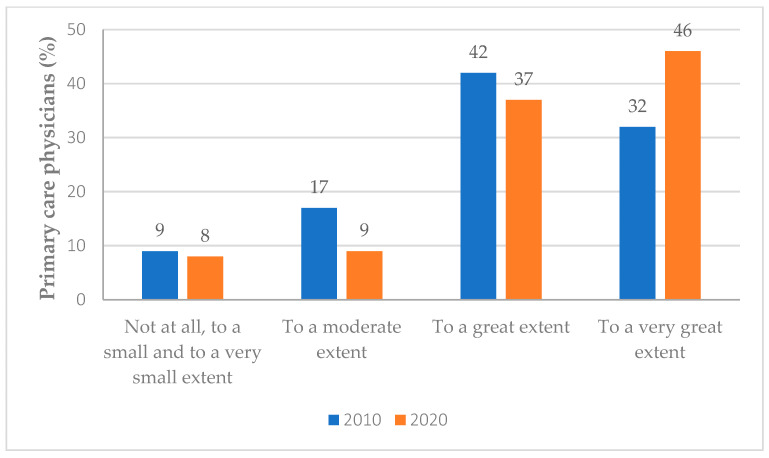
The extent of perceived shared responsibility of nurses by PCPs. Each Bar indicates weighted percentage of PCPs.

**Table 1 ijerph-19-10730-t001:** Demographic and work characteristics of the study population.

Variable	2010 N = 605 n (%)	2020 N = 450 n (%)	*p*-Value *
**Age**			<0.001
<45 years	157 (26)	76 (17)	
45–60 years	333 (55)	225 (50)	
>60 years	115 (19)	149 (33)	
**Sex**			NS
Female	226 (44)	180 (40)	
Male	339 (56)	270 (60)	
**Religion**			<0.001
Jewish	460 (76)	320 (71)	
Non-Jewish	145 (24)	130 (29)	
**Country of birth**			<0.005
Israel	242 (40)	220 (49)	
Other	363 (60)	230 (51)	
**Board certification**			NS
Family medicine	248 (41)	158 (35)	
Internist	121 (20)	99 (22)	
Non-specialist	236 (39)	193 (43)	
**Occupation**			<0.001
Primary care physician	557 (92)	396 (88)	
Specialist	48 (8)	54 (12)	
**Employment status**			NS
Employed by the health plan	284 (47)	189 (42)	
Self-employed (independent contractor)	170 (28)	171 (38)	
Both	151 (25)	90 (20)	

* *p*-value by chi-squared statistical test. NS = not significant.

**Table 2 ijerph-19-10730-t002:** Comparison of PCPs perceptions on nurses’ actual involvement contributes to the quality of practice by survey year (%).

Extent of Involvement	2010 Weighted % *	2020 Weighted % *
To a very great extent	17	25
To a great extent	42	42
To a moderate extent	26	21
To a small extent	9	7
To a very small extent and not at all	6	5

* Percentage reflect weighted data (see Section 2.4); *p* value < 0.05 for all comparisons.

**Table 3 ijerph-19-10730-t003:** Bivariate analysis of the association of the independent variables with physicians’ perceptions on nurses’ shared responsibility for improving quality-of-care by years of study.

Primary Care Physicians’ Response to Survey Items	Respondents Who Perceived That Nurses Share Responsibility for Quality-of-Care to a Very Great Extent Weighted Percentage * (*p*-Value)	
	2010	2020
**Psycho-social state affects medical condition and success of treatment**		
To a very great extent	43 (<0.001)	52 (<0.001)
**Follow-up on indicators increases work overload**		
To a very great extent	42 (<0.001)	46 (<0.001)
**Time devoted to follow-up and improvement of scores on indicators**		
Up to 5% of the time	34 (<0.001)	48 (<0.001)
**Nurses contribute to quality-of-care**		
To a very great extent	78 (<0.001)	86 (<0.001)
**Nurses share with me the performance measured**		
Definitely agree	**63** (<0.001)	70 (<0.001)
**I am responsible for the performance of some indicators and nurses—for others**		
Definitely agree	56 (<0.001)	64 (<0.001)

* The percentage reflects weighted data (see Materials and Methods). *p*-value by chi-squared test.

**Table 4 ijerph-19-10730-t004:** Logistic regression of physician perceptions of nurses’ shared responsibility for quality-of-care, to a very great extent.

Variable	Entered as	Reference Group	B	SE B	Odds Ratio	*p*-Value
Age, years	45–60	<45	−0.474	0.073	0.622	<0.001
	>60		−0.310	0.085	0.734	<0.001
Religion	Jewish	Non-Jewish	0.672	0.082	1.958	<0.001
Country of birth	Israel	Other countries	1.040	0.071	2.830	<0.001
Gender	Male	Female	−0.059	0.065	0.943	0.370
Board certification	Family physician	non-certified	−0.228	0.068	0.796	<0.001
	Internist/other		0.266	0.108	1.305	0.014
Form of employment	Salaried	Self-employed	0.198	0.079	1.219	0.012
	Both salaried and self-employed		0.261	0.089	1.298	0.003
Year of survey	2020	2010	0.205	0.077	1.228	0.007
Attitude to follow-up of quality-of-care	Increases work overload to a very high extent	Increases work overload to a low extent	0.180	0.065	1.197	0.005
	Commitment of up to 5% of time to indicators	Commitment of more than 5% of time to indicators	0.279	0.072	1.321	<0.001
Psycho-social state projects onto medical condition	To a very great extent	To a great extent or less	0.507	0.058	1.660	<0.001
Shared physician-nurse responsibility	Completely agree	Agree or less	0.321	0.068	1.379	<0.001
Full physician-nurse shared responsibility	Completely agree	Agree or less	1.600	0.064	4.954	<0.001
Nurses actually contribute to quality of practice	To a very great extent	To a great extent or less	1.970	0.077	7.173	<0.001
Cox and Snell R^2^					0.28	
Nagelkerke R^2^					0.34	
N					1055	

The regression included two dummy variables representing the different health plans and forms of questionnaire completion. All the included variables have a correlation coefficient no higher than 0.04 with respondents’ perceptions.

## Data Availability

Not applicable.

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
