# Peer review of "Primary Care Physicians’ Perceptions on Nurses’ Shared Responsibility for Quality of Patient Care: A Survey"

_ijerph, 2022, doi:10.3390/ijerph191710730_

Round 1
Author Response
Thank you for taking the time to review our manuscript and for the comments. Below please find a point-by-point response to each of the comments raised.
Comment 1: Before starting my referee, I state that I am not a doctor, nor a nurse and I am not Israeli and I do not live in Israel. So, I had to inform myself further, a situation (I think) that could happen to other readers of this journal. The purpose of the article is certainly interesting. However, it refers to a National Plan that can hardly be known by non-Israeli readers. In particular, I would suggest inserting in paragraph 2.1 a very brief explanation of these four planes to which reference is made (or clarify better, if they are already indicated).
Response: Thank you for this important comment. We have added to the introduction (lines 40-52) a brief explanation about the four health plans:
“Israel has a national health insurance system that provides universal coverage to all citizens and permanent residents of Israel. Each individual freely chooses from four competing, not-for-profit health plans (also referred to in the literature as health maintenance organizations, sick funds or health funds). These health plans provide their members with access to a statutory benefits package. The healthcare system is financed by general taxes and an earmarked payroll tax (health tax). These funds are allocated to the health plans according to a capitation formula, with risk adjustment intended to sufficiently compensate the plans for the cost of members' care. The members in each health plan choose their primary and specialist community-based physicians from physicians affiliated with the health plan [3,4]. The health plans vary in size, the characteristics of their members in terms of income, age and geographic location, the extent of reliance on salaried versus self-employed physicians, the extent of reliance on group practices versus solo practitioners, and whether the plan owns acute care hospitals [5].”
Comment 2: I would also suggest better explaining the phrase "Collected data were weighted to adjust for differences across plans in the number of employed nurses and response rates." (lines 104-105).
Response: We have clarified the sentence as follows (lines 149-156):” The data were weighted to reflect the differences among the health plans in sampling ratios and response rates (health plan-specific response rates ranged from 59% to 67%) so that the results would more accurately reflect the national study population. The weighting also considered the relationship between the sampling probability and the number of health plans where each physician worked (i.e., a physician working for two health plans was more likely to be included in the sample than a physician working for only one plan). All statistical analyses were conducted by using the weights.”
Comment 3: Some problems in understanding table 1 could depend on the previous point.
Looking at table 1, we should see that the two samples (of the 2010 and 2020 surveys) are different from each other for Age, Sex, Religion, Country of birth, Board certification, Occupation, Employment, Survey response mode; in fact, all comparisons report a significant probability.
But it is not clear what this probability refers to.
From the statement on lines 97-100, it would seem the use of a chi-square, but in the header of table 1 there is "n (%)" which should mean that there is a number and, in round brackets, the corresponding percentage. Instead, there is only a whole number which, however, appears to be a percentage (rounded or truncated to the integer). In fact, for each survey, each independent variable (Age, Sex, Religion,…) adds up to 100 except Age in 2010 (which adds up to 91, 25 + 47 + 19); in the articles [1,2] the data for Age are different (26 + 55 + 19 = 100), why? It is worth checking why 9% of statistical cases are missing. So, has the chi-square been calculated on the percentages, on the number of individual categories or on weighted numbers?
There is a further problem for which the accounts do not add up. If we look at the associated probability Sex ("<.003") and look at the data in the table, there is no theoretical correspondence. A 2x2 table, to have a significance chi-square, must show an interaction between the two variables. The fact that the percentages of males are always higher than that of females and the percentages of 2010 are always higher than those of 2020 leads me to expect an insignificant chi-square.
Trying to calculate the chi-square with the data reported in the table for Sex, an insignificant value is
Where does a p <.003 come from?
If the data are weighted, it would be better to report both the unweighted frequencies (which should add to 605 and 450) and the weighted ones and the percentages (of each of the two).
Reading the data in table 1, I expected that Religion and Occupation were not significant, while Country of birth, Survey response mode yes. Age is not significant with the percentages and significant with the frequencies (but the Contingency index is very low to indicate that the significance of the chi-square is due to the high number). So I would be careful; because as the total N increases (by weighing the statistical cases), the chi-square also increases.
Response: Following your comment the demographic data were analyzed again by a statistician and corrected. Comparisons were done by chi-squared test. And this information was added to the table. The differences in the variables between the survey years may be due to the time that had elapsed between surveys during which some of the population characteristics have changed.
Comment 4: In table 1, I would add, in addition to the probability, also the value of chi-quadro, the degrees of freedom, the effect size and (perhaps also) the power. But Spss 24 does not calculate it.
Response: In tables comparing demographic data in studies in public health policy, usually only the p value is presented. Additionally, SPSS 24 does provide output of the parameters mentioned.
Comment 5: On lines 121 and 128, there are references to figures A1 and A2, which are missing from the text.
Response: Figure 1 was accidently omitted and is now shown. We have decided to remove Figure 2 and instead show the data in a table (Table 2).
Comment 6: Table 2 and associated text (lines 133-144).
In the text or in the table, I suggest repeating that it is a chi-quadro with frequencies (or percentages) of weighed data and I would also report the standard information (chi, df, p, effect size) could be useful for a future meta- analyses.
Response: As we’ve added an additional table, the table referred to in the comment is now Table 3. The data were analyzed again by a statistician. Bivariate analysis of the association of the independent variables with PCPs’ perceptions on nurses’ shared responsibility for improving the quality-of-care were examined by study year by the chi-squared test.
Comment 7: Note that 2020 is written as "202".
If in table 1 you have put many horizontal lines, here some horizontal lines to separate each block, would simplify the reading.
Response: We have corrected the typo and have added horizontal lines to the table
Comment 8: Table 3 and associated text (lines 150-160).
I assume that even in the case of Table 3 the data used are weighted and that the logistic regression is binary with “very great extent” against “all other categories”. But what does the second-laste column contain? In theory it should be a standardized point (obtained with B divided by SE B), but… in this table it is not true. Furthermore, the same column, at the bottom, also contains an N and R2 (one even equal to 43!)
And what is the use of non -standardized parameters (B) and the relative standard error (SE B) if the reader has no idea what they mean (why are they weighed)? Wouldn't it be better to report a standardized estimate, which would allow us to understand the results shown in the text?
How can it be understood from Table 3 that "Logistic regression ... showed that Older Age, Male Gender, Self-Employed Status and Board Certification in family medicine were independent predictors of lower perception ..." If the first, third and fourth are significant but not the second?
Response: The analysis was reviewed again by a statistician. Indeed “43” was a typo. The correct value is 0.34.
The entire regression output is presented to the readers for them to obtain a complete picture of the analysis.
We have deleted “male gender” from the sentence in lines 222-227 and it now reads: “Logistic regression of data of PCPs who perceived that nurses share the responsibility for quality-of-care to a very great extent showed that older age, self-employed status and board certification in family medicine were predictors of lower perception regarding the extent of nurses’ responsibility for quality-of-care, while Israeli-born and Jewish PCPs were more likely to perceive nurses as sharing responsibility to a very great extent (Table 4).”
Comment 9: Table 4 and associated text (lines 167-174). What does "The study team conducted a series of multivariate models" mean? Multinomial logistics regressions or something different?
Response: Following a discussion with our statistician, we have omitted the table as it is irrelevant to the research question. This was a sensitivity analysis that does not answer the study question.
Reviewer 2 Report
The paper is well constructed from a theoretical point of view, however, the statistical presentation and analysis can be greatly improved.
For which question was the optimal sample size determined? This should be described more precisely.
How many people were asked to participate in the study at both measurement time points? What is the response rate here in terms of potential participants contacted? This is not clear to me when reading the paper.
For the duration of the data collection, the participation rate is quite low.
Did participants receive a benefit for participating (money or similar?).
In Table 1, decimal places should also be given for the percentage values. In addition, the specific N should always be shown.
Table 3: Nagelkerke cannot be 43. There seems to be a typo here. In the penultimate column of table is not designated, are the exp(B) values shown there?
The N for the logistic regression is really 605? Actually it is 1055?
As far as the content is concerned, I would like to point out that we are dealing here with self-judgments. It would be interesting to compare the values with the experiences of doctors, for example. At the same time, the survey also contains the problem of social desirability. This should also be included in the discussion as an important point.
Author Response
Thank you for taking the time to review our manuscript and for the comment. Below please find a point-by-point response to each of the comments raised.
Comment 1: The paper is well constructed from a theoretical point of view, however, the statistical presentation and analysis can be greatly improved. For which question was the optimal sample size determined? This should be described more precisely.
Response: We have added an explanation about how the sample size for the study was determined (lines 95-109): The study population consisted of PCPs working for the health plans (full- or part-time, salaried or self-employed contractors), engaged in the direct care of adult patients. Paediatricians, physicians with no responsibility for the quality of care for a panel of patients: consultants, physicians engaged mainly in administrative or managerial work, retired physicians, and temporary replacements were excluded from the study population. The study team estimated that approximately 4,400 Israeli physicians met these criteria. In each survey, a random stratified sample of 1,000 PCPs was selected from the administrative records of the health plans – 250 from each of the four health plans. In the 2010 survey, 884 PCPs met the eligibility criteria and were asked to participate in the survey, thereof 605 responded and completed the questionnaire (a response rate 68.4%). In the 2020 survey, 725 PCPs met the eligibility criteria, thereof, 450 completed the questionnaire (a response rate 62.0%). The main reasons for non-response were refusal to participate in the study and difficulty in making contact (18% and 20%, respectively, for the 2020 survey).”
Comment 2: How many people were asked to participate in the study at both measurement time points? What is the response rate here in terms of potential participants contacted? This is not clear to me when reading the paper.
Response: We have clarified the sentence regarding the number of people who were asked to participate in the study and those who responded (lines 102-109). “In each survey, a random stratified sample of 1,000 PCPs was selected from the administrative records of the health plans – 250 from each of the four health plans. In the 2010 survey, 884 PCPs met the eligibility criteria and were asked to participate in the survey, thereof 605 responded and completed the questionnaire (a response rate 68.4%). In the 2020 survey, 725 PCPs met the eligibility criteria, thereof, 450 completed the questionnaire (a response rate 62.0%). The main reasons for non-response were refusal to participate in the study and difficulty in making contact (18% and 20%, respectively, for the 2020 survey).”
Comment 3: For the duration of the data collection, the participation rate is quite low.
Response: The data were collected by cross-sectional survey at two time points: 2010 and 2020. A recent metanalysis that screened 8672 studies, and examined 1071 online survey response rates reported in education-related research showed that the average online survey response rate is 44.1% (Wu M-J, Zhao K, Fils-Aime F. Response rates of online surveys in published research: A meta-analysis, Computers in Human Behavior Reports. 2022;7. https://doi.org/10.1016/j.chbr.2022.100206). Comprehensive reviews examining survey response rates within primary care literature have reported response rates varying from 10.3% to 61% (Booker QS, Austin JD, Balasubramanian BA. Survey strategies to increase participant response rates in primary care research studies, Family Practice. 2021; 38(5): 699–702). According to the Pew Research Center, the response to telephone public opinion polls decreased from 36% in 1997 to 6% in 2018 (https://www.pewresearch.org/fact-tank/2019/02/27/response-rates-in-telephone-surveys-have-resumed-their-decline/).
Therefore, the response rates reported in our study are quite reasonable and in line with the known literature.
Comment 4: Did participants receive a benefit for participating (money or similar?).
Response: Participation was voluntary and participants did not receive a benefit for participating (money or similar?).
Comment 5: In Table 1, decimal places should also be given for the percentage values. In addition, the specific N should always be shown.
Response: We have added the specific N. Following a discussion with our statistician, it was decided to show the numbers as rounded up.
Comment 6: Table 3: Nagelkerke cannot be 43. There seems to be a typo here. In the penultimate column of table is not designated, are the exp(B) values shown there?
Response: We have reviewed Table 3 again with a statistician. Indeed “43” was a typo and it was corrected to 0.34.
Comment 7: The N for the logistic regression is really 605? Actually it is 1055?
Response: Thank you for noting this. Indeed the N is 1055.
Comment 8: As far as the content is concerned, I would like to point out that we are dealing here with self-judgments. It would be interesting to compare the values with the experiences of doctors, for example. At the same time, the survey also contains the problem of social desirability. This should also be included in the discussion as an important point.
Response: Thank you for this comment. We have added the issue of social desirability to the limitations of the study (lines 320-325): “The two surveys were conducted a decade apart, and although the questions about nurses’ contributions were identical, there was a difference in the method for completing the questionnaires: in the 2010 survey, 82% of questionnaires were completed by regular mail and in 2020 most questionnaires were completed online. These different means for survey completion may have led to a social desirability bias among respondents, despite the assurance of anonymity.”
Reviewer 3 Report
Comments and Suggestions for Authors
1. The list of references consists of 27 sources, the most recent from 2018. I suggest adding at least 5-6 articles from 2020 and newer to the list
2. I propose to expand the introduction with a short overview of analytical methods applied in public health and other medical studies. This would highlight the importance of choosing the statistical methods used in your manuscript from a wide range of other methods. I recommend the sources respectively:
- Subrahmanya et al. The role of data science in healthcare advancements: applications, benefits, and future prospects. Irish Journal of Medical Science (2021). https://doi.org/10.1007/s11845-021-02730-z
- Kraujalis, et al. Mortality rate estimation models for patients with a prostate cancer diagnosis. Baltic Journal of modern computing. 2022, vol. 10, no. 2, p. 170-184. DOI: 10.22364/bjmc.2022.10.2.06.
- Pain, et al. Deep learning-based image reconstruction and post-processing methods in positron emission tomography for low-dose imaging and resolution enhancement. European Journal of Nuclear Medicine and Molecular Imaging. 49, 3098–3118 (2022). https://doi.org/10.1007/s00259-022-05746-4
- Kim, et al. Relationships between nurses' experiences of workplace violence, emotional exhaustion, and patient safety. Journal of Research in Nursing. 2021 Mar;26(1-2):35-46. doi: 10.1177/1744987120960200.
3. Conclude the introduction with a paragraph about the scientific novelty of the article.
You have an extensive collection of interesting data. However, the analysis performed is very simple, and the obtained results answer the simple questions raised. The research would acquire significantly greater scientific value if the questions raised and the methods used to answer them were significantly more complex. Therefore, in the future, I recommend expanding the research team with a representative of statistics or data science who can perform serious multivariate data analysis. For example, perform clustering and create profiles of those clusters, perform exploratory and confirmatory factor analysis
Author Response
Response to Reviewer #3
Thank you for taking the time to review our manuscript and for the comment. Below please find a point-by-point response to each of the comments raised.
Comment 1: The list of references consists of 27 sources, the most recent from 2018. I suggest adding at least 5-6 articles from 2020 and newer to the list.
Response: Thank you for this suggestion. We have searched and added 6 newer relevant references.
- Lopes-Júnior, L.C. Advanced Practice Nursing and the Expansion of the Role of Nurses in Primary Health Care in the Americas. SAGE Open Nurs 2021, 7, 23779608211019491, doi:10.1177/23779608211019491.
- State of the World’s Nursing; World Health Organization: 2020.
- Swanson, M.; Wong, S.T.; Martin-Misener, R.; Browne, A.J. The role of registered nurses in primary care and public health collaboration: A scoping review. Nursing Open 2020, 7, 1197-1207, doi:https://doi.org/10.1002/nop2.496.
- Iriarte-Roteta, A.; Lopez-Dicastillo, O.; Mujika, A.; Ruiz-Zaldibar, C.; Hernantes, N.; Bermejo-Martins, E.; Pumar-Méndez, M.J. Nurses’ role in health promotion and prevention: A critical interpretive synthesis. Journal of Clinical Nursing 2020, 29, 3937-3949, doi:https://doi.org/10.1111/jocn.15441.
- Juraschek, S.P.; Zhang, X.; Ranganathan, V.; Lin, V.W. United States Registered Nurse Workforce Report Card and Shortage Forecast(). Am J Med Qual 2019, 34, 473-481, doi:10.1177/1062860619873217.
- Drennan, V.M.; Ross, F. Global nurse shortages-the facts, the impact and action for change. Br Med Bull 2019, 130, 25-37, doi:10.1093/bmb/ldz014.
Comment 2: I propose to expand the introduction with a short overview of analytical methods applied in public health and other medical studies. This would highlight the importance of choosing the statistical methods used in your manuscript from a wide range of other methods. I recommend the sources respectively:
Subrahmanya et al. The role of data science in healthcare advancements: applications, benefits, and future prospects. Irish Journal of Medical Science (2021). https://doi.org/10.1007/s11845-021-02730-z
Kraujalis, et al. Mortality rate estimation models for patients with a prostate cancer diagnosis. Baltic Journal of modern computing. 2022, vol. 10, no. 2, p. 170-184. DOI: 10.22364/bjmc.2022.10.2.06.
Pain, et al. Deep learning-based image reconstruction and post-processing methods in positron emission tomography for low-dose imaging and resolution enhancement. European Journal of Nuclear Medicine and Molecular Imaging. 49, 3098–3118 (2022). https://doi.org/10.1007/s00259-022-05746-4
Kim, et al. Relationships between nurses' experiences of workplace violence, emotional exhaustion, and patient safety. Journal of Research in Nursing. 2021 Mar;26(1-2):35-46. doi: 10.1177/1744987120960200.
Response: As the focus of this paper is how primary care physicians perceive nurses’ shared responsibility and contribution to quality-of-care, we think that reviewing statistical methods used in public healthcare and other medical studies would digress from the main focus of the manuscript and the introduction, which is meant to introduce the subject of the study. We agree that each field in health and medicine has its own methods and one does not always apply to the other. However, following your excellent suggestion, we intend to conduct a review of methods used in healthcare in the near future.
Comment 3: Conclude the introduction with a paragraph about the scientific novelty of the article.
Response: We have added a paragraph about the novelty of the article (lines 84-88): “Our findings will help in establishing policies to enhance the acceptance and appreciation for nurses’ contribution to quality-of-care, particularly among PCPs, and in the long-run should contribute to to attracting individuals to the nursing profession, especially in light of the ever-growing shortage of nurses [4,24,25].”
Comment 4: You have an extensive collection of interesting data. However, the analysis performed is very simple, and the obtained results answer the simple questions raised. The research would acquire significantly greater scientific value if the questions raised and the methods used to answer them were significantly more complex. Therefore, in the future, I recommend expanding the research team with a representative of statistics or data science who can perform serious multivariate data analysis. For example, perform clustering and create profiles of those clusters, perform exploratory and confirmatory factor analysis
Response: Thank you for this important suggestion. We will indeed include statisticians and representatives of data science in our next research.